# RBM24 in the Post-Transcriptional Regulation of Cancer Progression: Anti-Tumor or Pro-Tumor Activity?

**DOI:** 10.3390/cancers14071843

**Published:** 2022-04-06

**Authors:** De-Li Shi

**Affiliations:** 1Laboratory of Zebrafish Model for Development and Disease, Affiliated Hospital of Guangdong Medical University, Zhanjiang 524001, China; de-li.shi@upmc.fr; 2Laboratory of Developmental Biology, CNRS-UMR7622, Institut de Biologie Paris-Seine, Sorbonne University, 75005 Paris, France

**Keywords:** RBM24, RNA-binding protein, post-transcriptional regulation, cancer development, tumor progression, tumor suppressor, oncogene

## Abstract

**Simple Summary:**

RBM24 is a highly conserved RNA-binding protein that plays critical roles in the post-transcriptional regulation of gene expression for initiating cell differentiation during embryonic development and for maintaining tissue homeostasis in adult life. Evidence is now accumulating that it is frequently dysregulated across human cancers. Importantly, RBM24 may act as a tumor suppressor or as an oncogene in a context- or background-dependent manner. Its activity can be regulated by protein–protein interactions and post-translational modifications, making it a potential therapeutic target for cancer treatment. However, molecular mechanisms underlying its function in tumor growth and metastasis remain elusive. Further investigation will be necessary to better understand how its post-transcriptional regulatory activity is controlled and how it is implicated in tumor progression. This review provides a comprehensive analysis of recent findings on the implication of RBM24 in cancer and proposes future research directions to delve more deeply into the mechanisms underlying its tumor-suppressive function or oncogenic activity.

**Abstract:**

RNA-binding proteins are critical post-transcriptional regulators of gene expression. They are implicated in a wide range of physiological and pathological processes by modulating nearly every aspect of RNA metabolisms. Alterations in their expression and function disrupt tissue homeostasis and lead to the occurrence of various cancers. RBM24 is a highly conserved protein that binds to a large spectrum of target mRNAs and regulates many post-transcriptional events ranging from pre-mRNA splicing to mRNA stability, polyadenylation and translation. Studies using different animal models indicate that it plays an essential role in promoting cellular differentiation during organogenesis and tissue regeneration. Evidence is also accumulating that its dysregulation frequently occurs across human cancers. In several tissues, RBM24 clearly functions as a tumor suppressor, which is consistent with its inhibitory potential on cell proliferation. However, upregulation of RBM24 in other cancers appears to promote tumor growth. There is a possibility that RBM24 displays both anti-tumor and pro-tumor activities, which may be regulated in part through differential interactions with its protein partners and by its post-translational modifications. This makes it a potential biomarker for diagnosis and prognosis, as well as a therapeutic target for cancer treatment. The challenge remains to determine the post-transcriptional mechanisms by which RBM24 modulates gene expression and tumor progression in a context- or background-dependent manner. This review discusses recent findings on the potential function of RBM24 in tumorigenesis and provides future directions for better understanding its regulatory role in cancer cells.

## 1. Introduction

RNA-binding proteins (RBPs) are critically implicated in the post-transcriptional regulation of gene expression during physiological and pathological processes [1,2,3,4]. Through temporally and spatially regulated expression, dynamic shuttling in membrane- or phase-separated subcellular compartments, and combinatorial interactions with specific protein partners and RNA targets, they control the metabolism of RNAs at all stages of their lifetime, from pre-mRNA splicing and RNA editing to mRNA polyadenylation, transport, localization, stability/degradation and translation [1,5]. The human genome encodes more than 1500 verified RBPs with unique RNA-binding specificity and protein–protein interaction [6]. Increasing evidence suggests that they play essential roles in organizing the functional diversity of the proteome and in maintaining the homeostasis of protein synthesis during embryonic development and adult life. Because of their importance as key regulators of RNA biogenesis, dysfunctions of RBPs are closely associated with various human diseases [2,3,4]. In particular, a large number of RBPs are markedly dysregulated across human cancers, indicating that they are potentially involved in tumor development [7,8,9,10]. Moreover, there are also significant numbers of RBPs that are enriched for mutations in human cancers and are identified as candidate driver oncogenes [11]. Since RBPs constitute an intricate network regulating cell proliferation or differentiation, they could display either pro-tumor or anti-tumor activity [12]. Importantly, several RBPs may represent potential prognostic and diagnostic factors in cancer patients, but their contributions to tumorigenesis are largely underestimated [13]. Thus, a better understanding of the mechanisms by which RBPs function in cancer development should help to define therapeutic strategy for the treatment of various cancers [14,15,16]. Indeed, several approaches, such as small molecule inhibitors, antisense oligonucleotides, and peptides inhibiting protein–protein interactions, have been explored to target RBPs for cancer therapeutics [17].

The highly conserved RNA-binding motif protein 24 (RBM24) displays a strongly restricted tissue-specific expression pattern across vertebrate species [18,19], and functions as a multifaceted RBP in regulating cellular differentiation and maintaining tissue homeostasis [20]. It appears to be involved in many aspects of post-transcriptional regulation of gene expression during cell differentiation, including pre-mRNA splicing, mRNA polyadenylation and stability or translation [21,22,23,24,25,26]. These activities are likely dependent on its dynamic subcellular localization, biochemical interaction with specific protein partners and post-translational modifications [24,25,27]. Recently, there is increasing evidence showing that RBM24 expression is frequently dysregulated in human cancers. Functional and correlative analyses suggest that it displays either anti-oncogenic or oncogenic potential. Several studies suggest that it suppresses tumor progression [28,29,30,31,32], while others indicate that its upregulation promotes tumor growth [33,34,35,36,37]. These raise the possibility that RBM24 may act either as a tumor suppressor or as an oncogene, functioning in a context- or background-dependent manner. Nevertheless, it remains largely unclear how RBM24 exerts its anti-tumor activity and pro-tumor activity. Thus, further characterization of RBM24 expression and function across different cancers using clinical samples and appropriate animal models will be necessary to determine the mechanisms by which RBM24 modulates cancer development. This review discusses recent advances in studying the roles of RBM24 in cancer progression and proposes future directions in the research of RBM24-mediated post-transcriptional regulation of tumorigenesis.

## 2. RBM24-Mediated Post-Transcriptional Regulation of Gene Expression in Cancer Cells

Gene expression and function are also tightly controlled after transcription, with RBPs critically involved in various aspects of RNA biogenesis (Figure 1). Thus, post-transcriptional regulation makes an important contribution to proteomic diversity and homeostasis of protein abundances within a cell. RBM24 is an evolutionarily conserved RBP that regulates several post-transcriptional events in cancer cells (Figure 2). It contains a canonical RNA-recognition motif (RRM) at the N-terminal region, with two consensus ribonucleoprotein (RNP) sequences that bind AU/U-rich ligands present in a wide spectrum of target mRNAs [38]. The C-terminal half also comprises conserved motifs. In particular, there is a short region that interacts with eIF4E (eukaryote initiation factor 4E) and prevents it from binding to the 5′-cap of target mRNAs [24,39]. Phosphorylation of the serine residue present within this eIF4E-binding motif by GSK3β (glycogen synthase kinase 3ß) abolishes the interaction with eIF4E and converts RBM24 into an activator of mRNA translation [24,40]. For example, through binding to the U-rich element in the 3′-untranslated region (3′-UTR) of *p53* mRNA and interaction with eIF4E, RBM24 inhibits *p53* mRNA translation by preventing the assembly of translation initiation complex [24]. Besides this function in translation, RBM24 also regulates mRNA stability through binding to the 3′-UTR. In several cancer cell lines, such as MCF7 and HaCaT cells, RBM24 has been shown to increase the stability of *p21* mRNA, which encodes a cyclin-dependent kinase inhibitor [23]. Thus, it has the potential to induce cell cycle arrest and prevent tumor cell proliferation. There is also evidence that RBM24 destabilizes *p63* (*TP63*) mRNA by binding to its 3′-UTR [41]. The biological function of p63 transcription factor in cancers is complex because its two main isoforms, TAp63 and ΔNp63, exert opposite effects in tumorigenesis and metastasis. TAp63 displays tumor-suppressor features by inducing cell cycle arrest and cell death, while ΔNp63 exhibits oncogenic potential [42]. Thus, the outcome of RBM24-regulated *p63* mRNA stability on the expression TAp63 and ΔNp63 isoforms as well as the consequence on tumor development merit further investigation. Recent studies show that by binding to the 3′-UTR, RBM24 stabilizes *RUNX1T1* (RUNX1-related transcription factor 1) mRNA in bladder carcinoma and *PTEN* (phosphatase and tensin homolog) mRNA in colorectal cancer (CRC), with opposite effects on tumor progression [32,36].

In addition to mRNA targets, RBM24 also directly or indirectly regulates the expression of non-coding RNAs (ncRNAs) in cancer cells. In nasopharyngeal carcinoma (NPC) cell lines, analysis of microRNA (miRNA) expression profiles induced by RBM24 indicates that *miR-25* represents the most upregulated gene and mediates the tumor suppressor function of RBM24 [28]. However, the mechanism by which RBM24 increases *miR-25* expression is not clear. Given the important role of RBPs in the regulation of miRNAs during cancer initiation and progression [43,44], it will be of interest to further characterize RBM24-interacting ncRNAs across different cancers.

## 3. RBM24 in Cancer Development

### 3.1. RBM24 in Hepatocellular Carcinoma

Hepatocellular carcinoma (HCC) represents the most common form of liver cancer [45]. Large-scale analysis of transcriptome alterations indicates that RBM24 expression is frequently downregulated in HCC, which may trigger or sustain an undifferentiated state of tumor cells [7]. This suggests that RBM24 normally functions to inhibit HCC proliferation by promoting cell differentiation. Indeed, recent studies indicate that RBM24 shows reduced expression in liver cancer cell lines (HepG2, Hep3B, and Huh7) and exerts tumor-suppressive functions in HCC cells through several post-transcriptional mechanisms [30,31]. Overexpression of RBM24 prevents tumor cell growth and induces sorafenib sensitivity by indirectly reducing the expression level of *p63* mRNA likely through inhibition of β-catenin nuclear translocation [30]. Moreover, RBM24 can mediate the tumor suppressor function of ncRNAs by activating apoptotic tumor cell death. For example, the ncRNA *TPRG1-AS1* (tumor protein p63 regulated 1, antisense 1) has been shown to exert a tumor-suppressing property through stabilization of RBM24 expression by sequestrating its inhibitory *miR-3659* and *miR-4691-5p* [31]. These observations suggest that RBM24 displays potential tumor suppressor function in liver cancer. However, another study shows that RBM24 exhibits increased expression in HCC and can prevent the inhibitory effect of the E3 ubiquitin ligase TRIM56 (tripartite motif containing 56) on the proliferation of Huh7 and Bel-7402 cells, implying that it may exert an oncogenic function [33]. Further investigation will be necessary to clarify these apparent opposing results on RBM24 functions in HCC cells. In vivo functional assays, such as xenografts derived from RBM24-overexpressing or RBM24-deficient HCC cells combined with characterization of RBM24 target genes, should help to determine how it modulates HCC progression.

Chronic infection by hepatitis viruses represents the main risk factors for HCC development [46]. Several in vitro studies show that RBM24 can function as a host factor that interacts with RNAs of hepatitis B virus and hepatitis C virus to regulate their replication [47,48,49]. It appears that an appropriate level of RBM24 is required for hepatitis B virus replication in host cells, while either overexpression or knockdown of RBM24 can prevent virus replication [48]. These observations suggest that RBM24 may contribute to hepatitis virus infection and represent a potential target for developing antiviral strategies.

### 3.2. RBM24 in Gastrointestinal Cancers

Gastrointestinal cancers account for 26% of the total cancer incidence and 35% of all cancer-related mortality [50]. Colorectal cancer (CRC) represents the second most common cause of cancer death worldwide [51]. Dysregulation of several important signaling pathways including Wnt/β-catenin, Hedgehog, Notch, TGF-β and MAPK/PI3K are involved in gastrointestinal carcinogenesis [52]. Evidence is accumulating that RBM24 plays an important role in gastrointestinal cancers through different post-transcriptional mechanisms. One study demonstrates that RBM24 expression is strongly downregulated in colorectal tumor tissues from human patients and that spontaneous colorectal adenomas appear in *RBM24*-knockout mice [32]. This suggests that RBM24 has potential tumor suppressor function. Consistently, overexpression of RBM24 can inhibit metastasis of CRC cells and xenografts. Mechanistically, RBM24 promotes the accumulation of PTEN protein, a tumor suppressor and a negative regulator of PI3K (phosphoinositide 3-kinase) signaling, by directly binding to the 3′-UTR and increasing the stability of *PTEN* mRNA [32]. These results show that RBM24 can repress CRC progression by promoting the expression of tumor suppressor proteins. However, in gastric cancer, RBM24 expression seems to be associated with tumor cell migration or invasion [35]. Systematic profiling of splicing landscape of epithelial–mesenchymal transition (EMT) subtypes of gastric tumors indicates that RBM24 is upregulated in the tumor subtype displaying mesenchyme-specific alternative splicing, which is correlated with a poor prognosis in patients from The Cancer Genome Atlas (TCGA) clinical data [37]. In addition, the upregulation of RBM24 is positively correlated with the expression of the EMT activator ZEB1 (zinc finger E-box binding homeobox 1; previously known as TCF8), which impacts tumorigenesis by promoting migration, invasion, and metastasis [37,53]. This post-transcriptional activity is consistent with the function of RBM24 in alternative splicing during embryonic development [22]. However, the precise role of RBM24 in gastric cancer EMT remains unclear, and further identification of RBM24-regulated alternative splicing events will be necessary in order to understand how it modulates EMT in tumor progression and whether it exerts pro-tumor or anti-tumor activity in this cancer.

### 3.3. Tumor Suppressor Function of RBM24 in Nasopharyngeal Carcinoma

Expression profiling indicates that RBM24 and CELF4 are among the few RBPs that show frequent downregulation in nasopharyngeal carcinoma (NPC) tumor tissues and various NPC cell lines [28]. In functional assays, overexpression of RBM24 inhibits tumor growth in xenografts and suppresses migration and invasion of NPC cells, while knockdown of RBM24 produces the opposite effects, suggesting that it exerts anti-tumor activity in NPC [28]. Moreover, the tumor suppressor function of RBM24 appears to be mediated by *miR-25*, which is the most upregulated miRNAs in RBM24-induced cells and suppresses NPC progression by targeting the oncogenic lncRNA *MALAT1* (metastasis associated with lung adenocarcinoma transcript 1) for degradation [28]. These observations provide a mechanistic insight into the inhibitory role of RBM24 on NPC growth and viability. Further investigation will be necessary to analyze the clinical significance of RBM24 expression in NPC diagnosis and treatment. It is also of interest to determine how it functions in other types of head and neck cancers.

### 3.4. RBM24 Expression and Function in Lung Cancer

Lung cancer is one of the most common malignant tumors and a leading cause of cancer-related death worldwide [51]. Non-small cell lung cancer (NSCLC) represents a large majority of all lung cancer subtypes. It frequently displays advanced or distant metastasis at the time of diagnosis [54], which is associated with the process of EMT [55]. RBM24 may play a role in lung cancer development, but how it functions to modulate progression of this type of cancer remains unclear. There are conflicting reports on its expression and activity in NSCLC. One study shows that RBM24 exhibits reduced expression in NSCLC tissues. Functional assays using NSCLC cell lines suggest that it mediates the tumor suppressor activity of the circRNA *SMARCA5* to inhibit tumor growth and induce apoptosis [29]. By contrast, another study indicates that RBM24 protein shows increased expression in lung cancer tissues in a majority of cases, which may be correlated with a decreased chemosensitivity of lung adenocarcinoma (LUAD) cells and a reduced overall survival rate of patients with NSCLC [35]. Due to these discrepancies and the lack of detailed mechanistic analyses, the exact role of RBM24 in lung cancer awaits further functional investigation. Moreover, it is also important to identify RBM24-regulated targets and understand how it is involved in LUAD cell proliferation, migration and invasion. In this regard, it is worth mentioning that ZEB1 has been shown to play an important role in EMT and malignant progression [56]. Given the possible link between RBM24 and ZEB1 in gastric cancer [37], it will be also of interest to examine whether and how RBM24 modulates ZEB1 expression and EMT in lung cancer.

### 3.5. Pro-Tumor Activity of RBM24 in Bladder Cancer

RBM24 has been shown to display increased expression in bladder cancer tissues and appears to play an oncogenic role by promoting cell proliferation. Its upregulation is correlated with a poor prognosis in bladder cancer patients [36]. In addition, high expression of RBM24 appears to be associated with low overall survival and disease-free survival [57]. Functional analyses using human bladder carcinoma cells indicate that RBM24 stabilizes *RUNX1T1* mRNA and increases the expression of RUNX1T1 protein, which in turn positively regulates RBM24 expression by suppressing the transcription of its inhibitory *miR-625-5p* [36]. This positive feedback loop may represent a mechanism by which RBM24 participates in bladder cancer progression, but in vivo functional assays will be necessary to further determine its implication in tumor growth.

### 3.6. RBM24 in Other Cancers

RBM24 expression also appears to be dysregulated in several other cancer types, but functional and molecular analyses of its activity in tumor progression are still lacking. Bioinformatic analyses provide correlative evidence for a possible involvement of RBM24 in the development of several cancers. For example, survival analyses suggest that RBM24 may be a potential prognostic biomarker for head and neck squamous cell carcinoma (HNSCC) and skin cutaneous melanoma (SKCM) patients [58,59]. In triple negative breast cancer (TNBC) patients, RBM24 is upregulated during the disease-free interval (DFI) and is correlated with poor prognosis [34]. In medulloblastoma (MB), RBM24 represents one of the few reliable biomarkers that can be used for diagnosis and prognosis of group 4 tumors [60]. However, these correlative data require further experimental validation to determine how RBM24 regulates cell proliferation and tumor metastasis in these cancers. In particular, functional analyses using animal models associated with systematic identification of target genes will be necessary to understand its function in the development or recurrence of these cancers.

### 3.7. Possible Roles of RBM24 in the Progression of Other Diseases

RBM24 exhibits strongly restricted expression in skeletal muscle and in the heart during vertebrate development [18]. It also shows cytoplasm-to-nucleus translocation during myogenesis and dynamic post-transcriptional functions during muscle regeneration mediated by muscle stem cells [27]. In several animal models, loss of RBM24 activity has been shown to inhibit myogenic differentiation [18,61] and impair cardiogenesis [22,24,62,63,64] and vasculogenesis [62]. At present, no skeletal muscle disease has been directly associated with dysfunction of the *RBM24* gene. However, its altered expression has been observed in dystrophic muscular cells [65,66]. Of note, mouse models for muscular dystrophy are prone to myogenic tumors [67] and human patients with Duchenne muscular dystrophy present high susceptibility of developing rhabdomyosarcoma [68]. There is also evidence that muscle stem cells can give rise to rhabdomyosarcoma in dystrophic mice [69]. Thus, it will be of interest to examine whether dysregulation of RBM24 in muscle progenitor or stem cells impacts the occurrence of different myogenic tumors.

The most obvious phenotype resulting from loss of RBM24 is cardiogenic defects. RBM24-deficient mice develop dilated cardiomyopathy due to disrupted muscle-specific splicing of a large number of genes associated with cardiogenesis and sarcomere organization [22,64]. In addition, mutant embryos die at E13.5 due to endocardial cushion defect and growth retardation at least in part as a consequence of aberrant activation of p53-dependent apoptosis [24]. These observations suggest a possible involvement of RBM24 in heart disease and heart failure.

## 4. Regulation of RBM24 Expression and Activity in Cancers

It seems that RBM24 displays either decreased or increased expression as well as anti-tumor or pro-tumor activity, depending on the cancer type (Table 1). In several cancers, RBM24 functions as a tumor suppressor, and its low expression can lead to tumorigenesis. On the contrary, increased expression of RBM24 in other cancers may be associated with cell proliferation and tumor progression, resulting in poor prognosis and low overall survival. Thus, the challenge is to understand how the expression and activity of RBM24 are dysregulated during malignant transformation. There is evidence that the *RBM24* gene represents a transcriptional target of the tumor suppressor p53, which binds to *RBM24* promoter region and induces its expression in tumor cells independently of DNA damage [23]. The upregulation of RBM24 can, in turn, repress tumor progression by stabilizing mRNAs encoding tumor suppressor proteins. This is consistent with its anti-tumor activity in several cancer types. However, when the serine residue within the elF4E-binding motif is not phosphorylated, RBM24 can also inhibit p53 expression by interacting with elF4E and preventing the assembly of translation initiation complex [24]. In this situation, it may exert a potential pro-tumor function. These observations raise the intriguing possibility that RBM24 may have context-dependent activity and that the oncogenic or anti-oncogenic potential of RBM24 in different cancers may be modulated by phosphorylation. Accordingly, the elF4E-binding motif may represent a potential therapeutic target for cancer treatment. Indeed, disrupting the interaction between RBM24 and elF4E can effectively convert RBM24 into an activator of *p53* mRNA translation [24]. Therefore, it will be interesting to determine the phosphorylation status of RBM24 and identify possible kinases that are potentially involved in its post-translational modifications in different cancer tissues. In this regard, it is worth mentioning that GSK3β and Stk38 may play a role in RBM24 phosphorylation to regulate its function and stability [40,70].

Inappropriate epigenetic modifications can also contribute to dysregulation of RBM24 expression in cancers. It has been reported that cancer-related genes display differential methylation between tumor and normal tissues [71]. This mechanism can lead to overexpression of RBM24 in HCC cells and in HCC specimen. The reactivation of RBM24 expression is a consequence of increased demethylation in its enhancer but not in its promoter, and deletion of the enhancer region reduces its transcriptional level in Huh7 HCC cells but not in non-tumorigenic hepatocytes [71]. This raises a possibility that the decrease or increase in RBM24 expression during cancer progression may be dependent on its epigenetic modifications. There is also evidence that the function of RBM24 in cancers may be subjected to post-transcriptional regulation. As aforementioned, ncRNAs such as *TPRG1-AS1* and *circSMARCA5* can positively or negatively modulate the post-transcriptional expression of RBM24 in HCC and LUAD [29,31]. Altogether, RBM24 expression and function can be regulated at transcriptional, post-transcriptional and post-translational levels. Future studies will be necessary to provide further insights into the mechanisms underlying RBM24 dysregulation during malignant transformation and determine its activity in specific cancer cells.

## 5. RBM24-Interacting Proteins

Biochemical interactions of RBM24 with its protein partners are important for regulating different aspects of the post-transcriptional or post-translational process (Table 2). As discussed above, when it is not phosphorylated, the C-terminal eIF4E-binding motif of RBM24 and RBM38 (also known as RNPC1) interacts with eIF4E, while phosphorylation of the serine residue (position 181 in human RBM24 and position 195 in human RBM38) abolishes this interaction. Importantly, altered expression of eIF4E, often upregulated, has been also observed in different kinds of cancers [17]. Moreover, GSK3β appears to play a role in this phosphorylation and interacts with RBM38, especially its phosphorylated form [40]. Given the high degree of conservation between RBM24 and its paralog RBM38, it is likely that GSK3β similarly regulates the function of RBM24 on p53 expression. In myoblasts, there is evidence that RBM24 physically interacts with Stk38 through both the N- and C-terminal regions. Functionally, phosphorylation of RBM24 by Stk38 reduces the stability of RBM24 protein [70]. However, the priming substrate of Stk38 in RBM24 remains to be determined and whether this phosphorylation also occurs in cancer cells merits further investigation.

Because RBM24 is required for cytoplasmic polyadenylation to regulate cell differentiation during embryonic development, analysis of its potential protein partners in zebrafish shows that it interacts with cytoplasmic polyadenylation element-binding protein 4 (CPEB4) and cytoplasmic poly(A)-binding protein 1 (PABPC1), which regulate the elongation of poly(A) tail [25]. This suggests that RBM24 may be also a component of the cytoplasmic polyadenylation complex. The C-terminal region of RBM24 is required for this interaction, which may be also assisted by the N-terminal RRM [25]. How RBM24 functions in the cytoplasmic polyadenylation complex remains unclear. Its interaction with CPEB may play a role in poly(A) tail elongation, while its interaction with PABPC may regulate loop formation that brings the 5′- and 3′-ends of mRNA together or stabilize the poly(A) tail (Figure 3). It is well established that components of the cytoplasmic polyadenylation complex, in particular members of the CPEB family that are often upregulated in different cancers [17], are critically involved in regulating poly(A) tail length and mRNA translation during tumorigenesis and exert distinct activities in tumor progression [73,74]. For example, CPEB1 prevents breast cancer metastasis by reducing the poly(A) tail length and the expression of matrix metalloproteinase 9 [75], while CPEB4 promotes melanoma progression by modulating the expression of lineage-specific melanocyte-specific oncogenes, such as MITF (microphthalmia-associated transcription factor) and RAB7A, through cytoplasmic polyadenylation [76]. Thus, it will be interesting to examine how RBM24 functions with CPEB4 and PABPC1 to regulate the homeostasis of protein levels in both physiological and pathological processes.

RBM24 can also interact directly or indirectly with other closely related RBPs to coordinate cellular differentiation and function. For example, both RBM24 and RBM38 regulate *p21* and *p63* mRNA stability [23,41,77] and *p53* mRNA translation [24,78], raising the possibility they at least functionally interact in cancer cells. Physical and functional interactions between RBM24 and RBM38 in hepatitis B virus replication has been observed in HEK293T cells, which suggests that they form heterogeneous oligomers through their C-terminal regions [72]. Thus, deciphering the mechanism of RBM24-mediated protein–protein interaction in the post-transcriptional regulation of gene expression should help to define therapeutic strategies for cancer treatment.

## 6. Discussion

Evidence implicating RBM24 in malignant transformation is accumulating. Consistent with its differentiation-promoting function during embryonic development, RBM24 can act as a tumor suppressor to inhibit cell proliferation and tumor growth in several cancers. Nevertheless, there are also reports that correlate increased RBM24 expression with an oncogenic potential, but for many of these studies, in vivo functional assays combined with detailed mechanistic analyses are necessary to determine unequivocally the regulatory role of RBM24 in cancer development. It is possible that RBM24 exerts anti-tumor or pro-tumor activity in a context- or background-dependent manner. First, RBM24 can function either as a repressor or as an activator of *p53* mRNA translation, depending on the phosphorylation of the serine residue within the eIF4E-binding motif [24]. Thus, RBM24 phosphorylation can change the interaction with its protein partners and impact the regulation of its targets. In this regard, when considering dysregulation of RBM24 expression in cancer tissues, it may also need to be taken into consideration its phosphorylation status in addition to the analysis of its mRNA or protein levels. Second, RBM24 might regulate unique target genes in different kinds of cancers. For example, it stabilizes *PTEN* mRNA in CRC to prevent tumor growth [32] but increases *RUNX1T1* expression to drive bladder cancer progression [36]. Third, the activity of RBM24 in tumorigenesis may be also dependent on the expression of other cancer-related genes (genetic background). This background-dependent activity has been demonstrated for RBM38, a closely related RBM24 paralog. Loss of RBM38 decreases tumor penetrance when p53 is present [79], but this promotes tumor progression by reducing the expression of tumor suppressor genes when p53 is absent [80]. Thus, RBM38 can also function as a tumor repressor or as an oncogene, depending on the cancer types [17,81]. Indeed, RBM38 regulates similar targets as RBM24, and RBM38-deficient mice are also prone to spontaneous tumors [79]. Moreover, both RBM24 and RBM38 contain a C-terminal eIF4E-binding motif that regulates *p53* mRNA translation in a phosphorylation-dependent manner [24]. A synthetic peptide that disrupts RBM24/RBM38-eIF4E complex formation can induce p53 protein expression and suppress tumor development [39]. These works identify a critical role of the eIF4E-binding motif to modulate RBM24/RBM38 activity in cancer cells and raise the possibility that disrupting RBM24/RBM38-eIF4E complex may represent a potential therapeutic approach for cancer treatment.

RBM24 functions as a multifaceted post-transcriptional regulator of gene expression during cell differentiation [20]. Available evidence suggests that it suppresses or promotes cancer progression mostly through regulation of mRNA stability and translation. However, RBM24 also plays an important role in pre-mRNA splicing and mRNA polyadenylation, which are essential for cellular differentiation during organ development [22,25]. Whether and how RBM24 regulates these aspects of the post-transcriptional process in cancer cells remains largely elusive. Compared to embryonic development, only a few RBM24 targets have been identified in cancers. Thus, systematic profiling of RBM24-regulated events in various cancers will be of interest to understand its implication in the pathological processes. A related aspect is the lack of systematic examination of RBM24 subcellular localization in cancer cells. It has been shown that RBM24 displays dynamic cytoplasm to nucleus translocation during cellular differentiation, which is correlated with a shift from regulating mRNA stability to coordinating tissue-specific pre-mRNA splicing [27]. Importantly, data from the Human Protein Atlas indicate that RBM24 displays heterogeneous subcellular distribution in specific cancer tissues with a population of cells showing nuclear localization. While RBM24 is mostly accumulated in the nucleus of testis cancer cells, it is moderately present in the nucleus of a fraction of cells in some cases of lung, ovarian and prostate cancers, but is absent in the nucleus of other cancers. This suggests that it is differentially regulated and may exert distinct post-transcriptional activities in various cancers. Thus, it will be interesting to determine its cytoplasmic or nuclear localization in different cancer types and at different stages of cancer development to understand the multifaceted and dynamic features of its post-transcriptional regulatory functions. At present, only limited studies have been conducted to clarify the implication of RBM24 in a few cancer types. Future functional and mechanistic analyses across different cancers will be necessary to gain insights into its regulatory role in tumor progression. Moreover, deciphering the detailed mode of RNA-binding by the conserved RRM of RBM24 could provide a basis for generating engineered mutants to modulate its interactions with RNA targets [82]. They should also facilitate the discovery of potential therapeutic targets for cancer treatment.

## 7. Conclusions

Dysregulations of RBM24 are clearly associated with malignant transformation. Appropriate functional assays combined with in-depth molecular analyses should help to clarify whether and how it functions as a suppressor or as an oncogene in specific cancers. Its context- or background-dependent anti-tumor or pro-tumor activity is intriguing and merits further investigation. It appears that post-translational modifications of RBM24 play an important role in regulating its activity, but the mechanisms underlying this regulation remain largely elusive across different cancers with dysregulated RBM24 expression. Future studies using clinical samples and appropriate animal models will help to provide insight into the molecular and cellular mechanisms underlying RBM24 function in tumor progression. Importantly, targeting RBM24-mediated protein–protein interactions may also represent a potential therapeutic approach for cancer treatment.

## Figures and Tables

**Figure 1 cancers-14-01843-f001:**
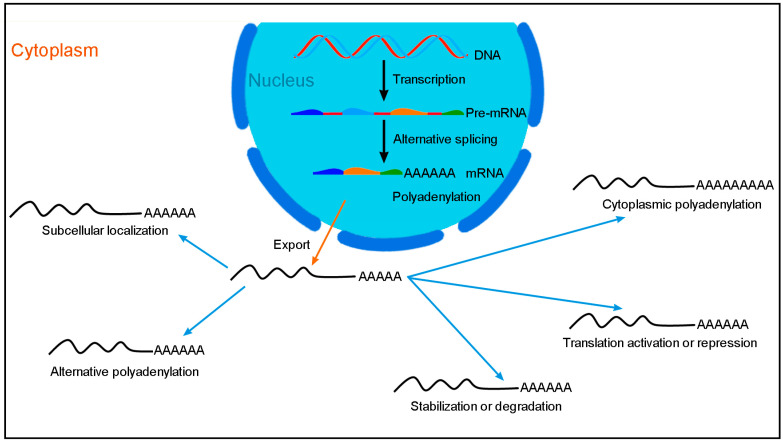
Post-transcriptional regulation of gene expression mediated by RBPs. In the nucleus, RBPs regulate alternative splicing and nuclear polyadenylation. Following export to the cytoplasm, mRNAs are subjected to multiple aspects of post-transcriptional regulation including restricted subcellular localization, alternative polyadenylation, cytoplasmic polyadenylation, translation activation or repression, and stabilization or degradation. All these processes not only increase the diversity of the proteome, but also maintain the homeostasis of protein synthesis and regulate the function of proteins within a cell.

**Figure 2 cancers-14-01843-f002:**
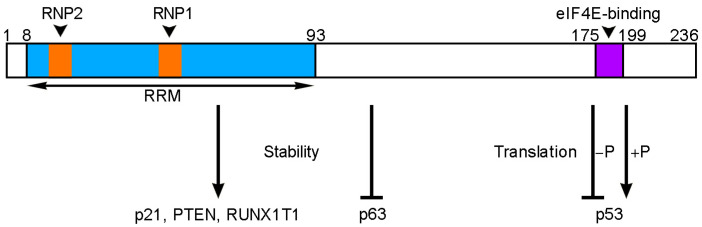
Conserved RBM24 functional domains and characterized RBM24 targets in cancer cells. Schematic shows human RBM24 protein with characterized functional motifs. Amino acid positions are indicated above. The N-terminal half contains a canonical RRM with two consensus RNP sequences (RNP1 and RNP2). The C-terminal region contains an eIF4E-binding motif that is involved in regulating the translation of *p53* mRNA, depending on the phosphorylation (−P or +P) of its serine residue. RBM24 also regulates the stability of *p21*, *PTEN*, *RUNX1T1* and *p63* mRNAs.

**Figure 3 cancers-14-01843-f003:**
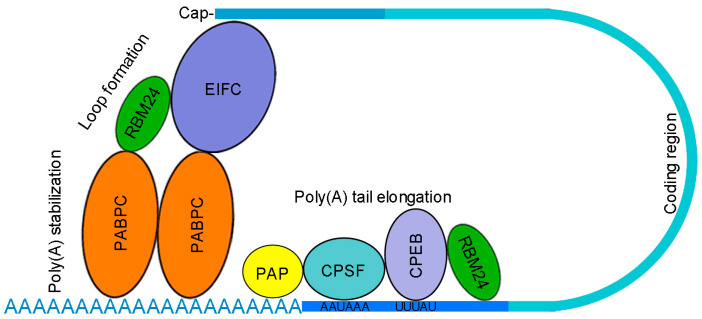
Putative functions of RBM24 in cytoplasmic polyadenylation. The interaction of RBM24 with CPEB may play a role in poly(A) tail elongation, while its interaction with PABPC may modulate loop formation or poly(A) stabilization. It remains to be determined how RBM24 physically interacts with different members of the CPEB and PABPC families. Functional assays will be also necessary to understand how RBM24 modulates the activity of CPEB and PABPC in cytoplasmic polyadenylation and cancer progression. EIFC, eukaryotic translation initiation factor complex; CPSF, cleavage and polyadenylation specificity factor; PAP, poly(A) polymerase.

**Table 1 cancers-14-01843-t001:** RBM24 expression, functions and targets in cancers.

Cancers	Expression	Activity or Outcome	Target Genes	Pro- or Anti-Tumor	Reference
HCC	Low	Suppress cell proliferation	β-catenin, p63	Anti-tumor	[30]
	Low	Inhibit cancer progression	Downstream of TPRG1-AS1	Anti-tumor	[31]
	High	Promote cell proliferation	Downstream of TRIM56	Pro-tumor	[33]
LUAD	Low	Inhibit tumor growth and induce apoptosis	Downstream of SMARCA5	Anti-tumor	[29]
	High	Decrease chemosensitivity	Unknown	Pro-tumor	[35]
CRC	Low	Repress tumor progression	PTEN	Anti-tumor	[32]
NPC	Low	Inhibit NPC growth and viability	miR-25	Anti-tumor	[28]
Bladder cancer	High	Increase tumor size	RUNX1T1	Pro-tumor	[36]
TNBC	High in DFI	Poor prognosis	Unknown	Pro-tumor	[34]
HNSCC	High	Low overall survival	Unknown	Pro-tumor	[58]
SKCM	High	Low overall survival	Unknown	Pro-tumor	[59]
MB (group 4)	High	Biomarker	Unknown	Pro-tumor	[60]

**Table 2 cancers-14-01843-t002:** RBM24-interacting proteins.

RBM24 Partners	RBM24 Regions Involved	Functions	References
eIF4E	eIF4E-binding motif	Prevent the assembly of translation initiation complex	[24]
GSK3β	eIF4E-binding motif	Phosphorylate the serine residue within the eIF4E-binding motif	[24,40]
Stk38	N- and C-terminal regions	Reduce RBM24 stability by phosphorylation	[70]
RBM38	C-terminal region	Regulate hepatitis B virus replication	[72]
CPEB4	C-terminal region	Cytoplasmic polyadenylation	[25]
PABPC1	C-terminal region	Cytoplasmic polyadenylation	[25]

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
