# Peer review of "RBM24 in the Post-Transcriptional Regulation of Cancer Progression: Anti-Tumor or Pro-Tumor Activity?"

_cancers, 2022, doi:10.3390/cancers14071843_

Round 1

Reviewer 1 Report

De-Li Shi described the implication of RNA-binding protein RBM24 in gene expression, epigenetic modification and tumor progression. The author provides a future directions for the detection of mechanisms underlying tumor suppressor function or oncogenic activity of RBM24. The manuscript covers several different aspects and is well organized. It is an update.

Author Response

De-Li Shi described the implication of RNA-binding protein RBM24 in gene expression, epigenetic modification and tumor progression. The author provides a future directions for the detection of mechanisms underlying tumor suppressor function or oncogenic activity of RBM24. The manuscript covers several different aspects and is well organized. It is an update.

Thank you for your positive assessment of this work.

Reviewer 2 Report

The manuscript provides a detailed description of the involvement of RBM24 in cancer-related processes. The topic is interesting, as RNA binding proteins and non-coding RNAs are getting more and more recognition as important regulators of cellular and pathological processes. Even though the available information on the function and relevance of RBM24 is limited, a systematised description can help researchers in future discoveries.

The manuscript is overall well written, even though some grammatical errors can be found in the text. The information provided is clearly categorised and is presented in an easy-to-follow manner.

I only have some minor comments:

  1. The yellow background for the figures is unnecessary and a little hard on the eyes, I suggest presenting the figures with a plain white background.
  2. Table 1.: For the convenience of the readers, it might be useful to include an extra column that states whether the given activity is considered pro- or anti-cancer. It can of course be deduced from the function, but it would make the interpretation of the information easier.
    Also, regarding this table: interestingly, if one considers exclusively the information in the table, it shows that low expression of RBM24 correlates with tumor suppression and high expression leads to tumor progression and poor prognosis. This would suggest a clear oncogene function, in contrast to the ambiguous information detailed in the text. I suggest to address this apparent contradiction in some manner.
  3. Grammatical errors and typos I found:
    Line 162: displays instead of display
    Line 215: "which is a mostly upregulated miRNAs" - might be "which is the most unregulated miRNA".
    Line 244: been instead of bee
    Line 375: "it may also need to take into consideration"  - "it may also need to be taken...
    There might other mistakes be in the text I haven't noticed, so I suggest a thorough spelling and grammar check.

Author Response

1. The yellow background for the figures is unnecessary and a little hard on the eyes, I suggest presenting the figures with a plain white background.

I followed this suggestion by presenting all figures with a plain white background.

2. Table 1.: For the convenience of the readers, it might be useful to include an extra column that states whether the given activity is considered pro- or anti-cancer. It can of course be deduced from the function, but it would make the interpretation of the information easier.
Also, regarding this table: interestingly, if one considers exclusively the information in the table, it shows that low expression of RBM24 correlates with tumor suppression and high expression leads to tumor progression and poor prognosis. This would suggest a clear oncogene function, in contrast to the ambiguous information detailed in the text. I suggest to address this apparent contradiction in some manner.

Thanks for the suggestions and comments.

For the first point, an extra column was added to Table 1 indicating pro- or anti-tumor activity of Rbm24 in different cancers.

The second point was addressed by the following sentences: “In several cancers, RBM24 functions as a tumor suppressor, and its low expression can lead to tumorigenesis. On the contrary, increased expression of RBM24 in other cancers may be associated with cell proliferation and tumor progression, resulting in poor prognosis and low overall survival”.

3. Grammatical errors and typos I found:
Line 162: displays instead of display

It is corrected.

Line 215: "which is a mostly upregulated miRNAs" - might be "which is the most unregulated miRNA".

This is changed.

Line 244: been instead of bee

It is corrected.

Line 375: "it may also need to take into consideration"  - "it may also need to be taken...

This is corrected.

There might other mistakes be in the text I haven't noticed, so I suggest a thorough spelling and grammar check.

Thanks for this recommendation. Thorough spelling and grammar check was done during the revision.

Reviewer 3 Report

The review: „RBM24 in the Post-Transcriptional Regulation of Cancer Progression: Anti-Tumor or Pro-Tumor Activity? " by De-Li Shi elaborately discusses recent findings on the potential function of RBM24 in tumorigenesis and provides future directions for better understanding its regulatory role in cancer cells. RNA-binding proteins are known to control the metabolism of RNAs throughout their lifetime and are invariably required for the post-transcriptional regulation of gene expression in a lot of physiological and pathological processes. Among other functions, the author discusses an association of malignant transformation with dysregulations of RBM24. The review is nicely composed with citation of a number of relevant literatures. After thoroughly going through the manuscript, I have a couple of comments:

  1. Despite association of RBM24 in malignant transformation, the Rbm24 gene is reported to be expressed at all sites of skeletal muscle development as well. Although, the submitted manuscript invariably focuses in anti- or pro tumor activities of RBM24, I would like to suggest the author to briefly discuss on role of RBM24 in myogenic differentiation during vertebrate development and its correlation with tumor activity.
  2. Mechanism of RBM24 modulated cancer development through animal models has not been analyzed to date. However, a few animal model studies reporting loss of RBM24 activity in disease progression in different diseases (such as acquired heart disease) are known. Please highlight this point also in the manuscript.

Author Response

1. Despite association of RBM24 in malignant transformation, the Rbm24 gene is reported to be expressed at all sites of skeletal muscle development as well. Although, the submitted manuscript invariably focuses in anti- or pro tumor activities of RBM24, I would like to suggest the author to briefly discuss on role of RBM24 in myogenic differentiation during vertebrate development and its correlation with tumor activity.

Thanks for the suggestion. The revised manuscript includes an additional section “Possible Roles of RBM24 in the Progression of Other Diseases” (section 3.7) discussing RBM24 function in myogenic differentiation and its possible link with myogenic tumors, such as rhabdomyosarcoma.

2. Mechanism of RBM24 modulated cancer development through animal models has not been analyzed to date. However, a few animal model studies reporting loss of RBM24 activity in disease progression in different diseases (such as acquired heart disease) are known. Please highlight this point also in the manuscript.

I followed this suggestion by highlighting RBM24 function in heart development and its potential implication in dilated cardiomyopathy, in the same section as above.